# Soft Pneumatic Muscles: Revolutionizing Human Assistive Devices with Geometric Design and Intelligent Control

**DOI:** 10.3390/mi14071431

**Published:** 2023-07-16

**Authors:** Mahmoud Elsamanty, Mohamed A. Hassaan, Mostafa Orban, Kai Guo, Hongbo Yang, Saber Abdrabbo, Mohamed Selmy

**Affiliations:** 1Mechanical Department, Faculty of Engineering at Shoubra, Benha University, Cairo 11672, Egypt; 2Mechatronics and Robotics Department, School of Innovative Design Engineering, Egypt-Japan University of Science and Technology, Alexandria 21934, Egypt; 3School of Biomedical Engineering (Suzhou), Division of Life Sciences and Medicine, University of Science and Technology of China, Hefei 230026, China; 4Suzhou Institute of Biomedical Engineering and Technology, Chinese Academy of Sciences, Suzhou 215163, China; 5Electrical Department, Faculty of Engineering at Shoubra, Benha University, Cairo 11672, Egypt; mohamed.selmy@feng.bu.edu.eg

**Keywords:** soft actuator, continuum robots, soft muscle, soft pneumatic muscle, soft parallel muscles, FEA, artificial neural network

## Abstract

Soft robotics, a recent advancement in robotics systems, distinguishes itself by utilizing soft and flexible materials like silicon rubber, prioritizing safety during human interaction, and excelling in handling complex or delicate objects. Soft pneumatic actuators, a prevalent type of soft robot, are the focus of this paper. A new geometrical parameter for soft artificial pneumatic muscles is introduced, enabling the prediction of actuation behavior using analytical models based on specific design parameters. The study investigated the impact of the chamber pitch parameter and actuation conditions on the deformation direction and internal stress of three tested soft pneumatic muscle (SPM) models. Simulation involved the modeling of hyperelastic materials using finite element analysis. Additionally, an artificial neural network (ANN) was employed to predict pressure values in three chambers at desired Cartesian positions. The trained ANN model demonstrated exceptional performance. It achieved high accuracy with training, validation, and testing residuals of 99.58%, 99.89%, and 99.79%, respectively. During the validation simulations and neural network results, the maximum errors in the x, y, and z coordinates were found to be 9.3%, 7.83%, and 8.8%, respectively. These results highlight the successful performance and efficacy of the trained ANN model in accurately predicting pressure values for the desired positions in the soft pneumatic muscles.

## 1. Introduction

Significant advancements in soft pneumatic actuators, commonly known as pneumatic artificial muscles (PAMs), have led to notable progress in soft robotics. The origins of soft robotic muscles can be traced back to the 1950s when McKibben actuators were introduced as linear actuators, laying the foundation for the subsequent development of pneumatic muscles [1]. A significant milestone in this domain was achieved with the encasing of textile threads within a cylindrical rubber tube, enabling the creation of flexible and versatile actuation mechanisms [2]. The introduction of elastomer-based actuators further contributed to the evolution of soft robotics, showcasing their potential for compliant and biomimetic movements [3]. Soft robotics has garnered significant enthusiasm due to its potential to address challenges across various domains, including healthcare, exploration, and human–robot collaboration [4,5]. Soft robotic systems offer inherent compliance, providing advantages for safety, adaptability, and dexterity. These qualities make soft robots well-suited for applications where conventional rigid robots may have limitations or pose risks. While conventional robotics has laid a solid foundation for technological advancements, the emergence of soft robotics presents exciting opportunities to explore new frontiers in robotics research and development. The dynamic and evolving nature of soft robotics holds promise for transforming industries and enhancing the capabilities of robotic systems in unprecedented ways.

Pneumatic artificial muscles (PAMs) have emerged as a powerful technology in soft robotics, playing a crucial role in assistive muscle and upper limb rehabilitation applications. PAMs possess unique properties that enable safe and seamless interactions between humans and robots, making them highly suitable for performing intricate or delicate tasks. These versatile actuators find applications in diverse domains, including mobile robots, wearable devices, and exoskeletons, contributing to realizing assistive and rehabilitative functionalities [3,4]. Leveraging the inherent capabilities of PAMs, soft robotics continues to revolutionize various sectors, offering unprecedented possibilities for advanced robotic systems characterized by exceptional functionality, safety, and adaptability. Pneumatic artificial muscles, also known as PAMs, are flexible actuators specifically designed to mimic the movement of human muscles. Their significance in robotics and medicine is underscored by ongoing research efforts to develop lightweight and compact PAMs with high power-to-weight ratios. A comprehensive review [6] provides an overview of various types of PAMs, their manufacturing processes, force models, and diverse applications in robotics, medicine, and industry. The review highlights the crucial aspect of enhancing muscle dynamics to achieve precise control, emphasizing the need for further advancements in this area.

## 2. Advancements in Soft Pneumatic Muscles

A seminal study [7,8] presents a significant advancement in pneumatic actuation, introducing the development of the first self-sensing inverse pneumatic artificial muscle. This groundbreaking actuator surpasses the capabilities of conventional positive pneumatic actuators by offering an expanded range of motion. It incorporates a reinforcing yet compliant element that effectively guides the actuator’s movement while enabling self-sensing through its electrical conductivity. The research outlines the actuator’s design, manufacturing process, and comprehensive electromechanical characterization, elucidating its remarkable self-sensing capabilities in dynamic operational scenarios. In a complementary investigation [9,10], the focus centers on pneumatic muscle fibers (PMFs) and their functionality as pliable actuators when subjected to compressed fluid within their bladder. By employing a nonlinear quasi-static model, the study comprehensively examines the stiffness and variable stiffness characteristics of pressurized PMFs. The experimental findings robustly validate the model’s predictions, reaffirming its efficacy. Furthermore, the study presents a compelling demonstration of a composite material embedded with PMFs, effectively showcasing its potential for facilitating adaptive shape-changing behaviors and flexible stiffness adjustment.

In contrast to the extensive field of conventional robotics, soft robotics has emerged as a distinct and promising research domain, reshaping the landscape of robot design and motion. Soft robotics represents a paradigm shift from traditional approaches, as it leverages the inherent properties of its actuators to achieve motion without relying on external torque or conventional motors [11,12,13,14]. Soft robots employ soft and flexible materials such as elastomers and silicone rubber, enabling versatile and adaptable movements. This innovative approach facilitates interactions with complex and delicate objects and navigation in unstructured environments. The absence of rigid components and intricate linkages simplifies the design process and enhances the safety of human–robot interactions [15]. The potential of soft robotics in assistive muscle and upper limb rehabilitation is particularly promising, offering precise and controlled assistance to recover and improve upper limb functionality [16,17]. Many soft robotics designs take inspiration from biological organisms, aiming to replicate and enhance their movements for research and technological advancements [18,19]. These bio-inspired approaches leverage the inherent behavioral characteristics of biological creatures, enabling soft robots to interact with objects and their environment without relying on complex control systems [8,20]. Unlike conventional robots that require precise kinematic models for motion planning, soft robots exhibit a unique linguistic behavior that allows them to adapt and respond to their surroundings. While conventional robotics benefits from well-established kinematic models, the same cannot be applied to soft robotics. Soft robotic systems’ intrinsic flexibility and deformability make it challenging to capture their unique characteristics using traditional kinematic models. Consequently, there is a pressing need to develop specific kinematic models tailored to the requirements of soft robotics. These models should account for the soft and compliant nature of the robots, enabling accurate prediction and control of their movements [21,22].

The deformation behavior of soft robots is influenced by various factors, including the selection of materials [21,23,24], fabrication techniques [25,26], and chamber design [5,25,27,28]. Choosing materials with appropriate mechanical properties is critical in achieving the desired deformations and ensuring the optimal performance of soft robotic systems. Mechanical properties such as elasticity and stiffness directly impact the deformability and response of the soft robot. Moreover, the fabrication techniques employed significantly determine soft robots’ overall structure and functionality. Techniques such as 3D printing, molding, and casting enable the creation of intricate and customized soft robot structures. The design of chambers or compartments within the soft robot, which houses the actuators and control mechanisms, also contributes to the deformation behavior. Factors such as these chambers’ size, shape, and arrangement substantially influence the soft robot’s overall performance and capabilities. Comprehensive understanding and accurate modeling of the complex interplay among these factors is of utmost importance in advancing the field of soft robotics, particularly within rehabilitation applications. Through the meticulous investigation and optimization of material properties, fabrication techniques, and chamber designs, researchers can greatly enhance soft robotic systems’ performance, dexterity, and capabilities, specifically focusing on artificial assistive soft pneumatic muscles [29,30]. These advancements can potentially revolutionize the healthcare industry by enabling the development of robots that can safely and effectively support and assist individuals in their rehabilitation journey. Furthermore, developing precise and reliable kinematic models tailored to soft robotics will facilitate the precise control and coordination of movements, thereby enabling personalized and targeted rehabilitation interventions. This progress expands the range of applications and potential benefits of soft robotic systems in rehabilitation.

Soft pneumatic muscles, also known as flexible artificial pneumatic actuators, are designed to mimic the actuation properties of human muscles [31,32]. These muscle actuators, constructed using soft and pliable materials to ensure safe user interactions, offer variable compliance and have diverse applications in robotic systems and medical contexts [33]. These actuators are crucial in rehabilitation in physical therapy sessions [34,35]. They help alleviate fatigue and assist in rehabilitating individuals recovering from strokes by supporting their motion, managing workloads, and reducing the risk of injuries [36]. The soft muscle design, combined with fabric construction, enables significant deformation lengths, making it suitable for replicating the entire range of motion of the upper limb in rehabilitation scenarios [37]. The SPM distinguishes itself from other soft actuators by utilizing air pressure to induce deformations [38]. These artificial muscles can consist of one or more chambers within a single unit, offering a versatile range of motions, including linear extension, bending, contraction, and twisting actuation [33,39]. Smooth pneumatic muscles find applications in various domains, with a special focus on assistive and rehabilitation contexts [40,41]. An example of a wearable robotic device designed for ankle–foot rehabilitation emulates the musculoskeletal system of the human foot, providing active assistance while preserving natural ankle joint freedom [42]. Equipped with four pneumatic artificial muscles and gait analysis sensors, this device achieves a range of motion of 27 degrees. Experimental results validate its controllability using a linear time-invariant (LTI) controller [43]. Another novel actuator design, the folded pneumatic artificial muscle (foldPAM), features symmetrically folded, thin-filmed air pouches. This design enables closed-loop position control with adjustable force–strain relationships while maintaining constant pressure. Experimental results demonstrate precise closed-loop geometry control within a 0.5% error range [44].

Significant advancements have been made in the field of fluidic actuators, contributing to the development of innovative solutions for various applications in robotics and automation. One noteworthy example is the bidirectional electrohydrodynamic pump (BEDP), which has gained attention due to its high symmetrical performance and its potential application in tube actuators. The BEDP utilizes the principles of electrohydrodynamics, employing an electric field to induce fluid flow in a conductive medium. This technology enables precise control and efficient actuation, making it a promising candidate for integration into artificial pneumatic muscle systems for human assistive devices [45]. Another notable development in fluidic actuators is the emergence of the fluidic rolling robot, which utilizes voltage-driven oscillating liquid to achieve locomotion. By manipulating the fluid within its body, the robot is capable of smooth and efficient rolling motion [46]. The utilization of voltage-driven oscillating liquid as a driving mechanism offers advantages such as simplicity, scalability, and potential for miniaturization, presenting opportunities for advancement in the field of artificial pneumatic muscles.

A proposed strategy, referred to as echo-state Gaussian process-based nonlinear model predictive control (ESGP-NMPC), addresses challenges related to trajectory tracking using pneumatic muscle actuators (PMAs) in rehabilitation robots [42,43,44]. This strategy incorporates ESGP for uncertainty modeling and utilizes a gradient descent optimization algorithm, resulting in superior model fitting and control performance compared to conventional methods. Both simulation and physical experiments validate the effectiveness of this approach in achieving high-precision tracking tasks. Continually exploring and understanding the intricate relationships among material properties, fabrication techniques, and chamber designs play a crucial role in advancing soft robotics, particularly in assistive and rehabilitation muscles. By continually refining and optimizing these factors, researchers can enhance the capabilities and versatility of soft robotic systems, leading to groundbreaking advancements in healthcare, exploration, and human–robot interaction. The interdisciplinary nature of soft robotics fosters collaborations across diverse fields, facilitating the development of innovative solutions. Incorporating soft pneumatic muscles, with their safe and compliant actuation properties, proves instrumental in realizing effective and reliable assistive and rehabilitation applications.

This study aims to design and determine the geometrical parameters for the SPM and soft parallel muscle, model the soft actuators using finite element analysis (FEA), and investigate the influence of varying the parameter P with sequential values of pressures. Additionally, the study focuses on applying these soft muscles for rehabilitation purposes. Furthermore, an (ANN) is employed to establish both forward and inverse kinematics for the SPM and soft parallel muscle models, enabling the estimation of pressure values given the endpoint Cartesian coordinates (X, Y, and Z). The manuscript’s structure is organized as follows: Section 4 describes the model design of the SPM and soft parallel muscle, including the geometrical parameters of the soft muscles. Subsequently, Section 5 focuses on the finite element analysis, presenting the results and analysis conducted under different pressure conditions. Section 6 introduces a neural network regression model, providing an evaluation of the performance of the SPM and soft parallel muscle. Finally, Section 7 concludes the study with a brief discussion of the findings.

## 3. Upper Limb Assistive Device

Human assistive devices play a vital role in enhancing the quality of life for individuals with physical disabilities or impairments. These devices encompass a broad range of technologies designed to aid individuals in overcoming mobility challenges, improving functional capabilities, and promoting independence. Soft pneumatic muscles, as described in our study, have promising applications in the field of human assistive devices for upper limb support due to their inherent flexibility, adaptability, and safety characteristics.

One prominent application of soft pneumatic muscles is in the development of wearable exoskeletons for upper limb assistance. These devices provide external support and assistance to the wearer’s arms and hands, helping them perform daily tasks that may have been difficult or impossible due to physical limitations. Soft pneumatic muscles can be integrated into upper limb exoskeletons in order to provide a more natural and compliant interaction between the device and the human body, reducing the risk of injury and discomfort while improving overall functionality.

The proposed soft muscle will be suitable to fit a wearable exoskeleton design for the shoulder joint and incorporates three parallel soft pneumatic muscles, strategically arranged to enable efficient actuation and a wide range of motion. Each muscle is designed to be identical and consists of three chambers, allowing for tailored pressure application and control in each chamber, as shown in Figure 1. This configuration facilitates precise and smooth motion in multiple movement spaces, such as flexion extension, abduction–adduction, and internal–external rotation, closely mimicking the biomechanics of a healthy human shoulder. The soft pneumatic muscles are constructed using a flexible elastomeric material, such as silicone rubber, providing a compliant and comfortable interface with the user’s body while reducing the risk of injury or discomfort.

In addition to the shoulder actuation, another SPM is integrated into the design to operate and assist the human forearm and enable elbow joint rotation. This single-chamber muscle, also constructed from flexible elastomeric material, can be pressurized to induce controlled bending and extension of the elbow joint, effectively assisting the user in daily tasks that require forearm movement and manipulation. The muscle’s geometry and pressure control can be optimized to provide the desired range of motion, force output, and responsiveness, considering the user’s specific needs and physical abilities.

The combination of the three parallel soft pneumatic muscles for shoulder actuation and the additional muscle for elbow joint rotation presents a comprehensive solution for upper limb assistance. This soft wearable exoskeleton design offers a more natural and intuitive interaction between the device and the human body, closely replicating the complex biomechanics of the upper limb. By incorporating soft pneumatic muscles into the exoskeleton, users can benefit from enhanced mobility, improved force output, and a greater sense of embodiment, ultimately leading to a better user experience and satisfaction with the assistive device.

## 4. Model Design

Soft pneumatic muscles are regarded as attractive for soft robotics applications due to their flexible and inflatable nature, enabling a wide range of motions and forces. Typically, a SPMdesign consists of three parallel muscles, thereby facilitating stimulating motion in multiple movement spaces. By incorporating a model with more than one chamber, achieving larger workspace movements for actuated muscles becomes possible. This study devised a three-chamber model, with each chamber designed to be identical. The selection of geometrical parameters was undertaken following a rigorous benchmarking process to ensure the optimal performance of the soft pneumatic muscles. The resulting model exhibits motions with potential applications in post-stroke patient rehabilitation and physical therapy. An in-depth analysis was conducted to examine the motion under different conditions at predetermined air pressure values. The geometrical parameters utilized in the models are visually represented in Figure 2, providing a comprehensive depiction of the key design attributes.

The soft artificial pneumatic muscle (SPM) model comprises three chambers constructed from a flexible elastomeric material, specifically silicone rubber, and utilizes chamber rings instead of a reinforcement layer. These chambers are pressurized with air or fluid to induce controlled muscle inflation and prevent unintended expansion. The relevant muscle parameters, including the length of the chamber (L), outer diameter (Do), the height of the chamber (H), pitch (P), and chamber thickness (T), with a constant number of rings, are presented in Table 1. The chamber thickness is 7 mm, employing a flexible silicone rubber material capable of expansion, bending, and contraction in response to pressure changes. Pressurizing the muscle causes extension, while soft silicon rubber end caps are employed. Inlet ports facilitate fluid connection, enabling the muscle to be inflated or deflated. Considerations such as the desired range of motion, force output, and durability are crucial when designing a soft pneumatic muscle. These factors dictate material selection, size, shape, and the proper air pressure and flow rate for muscle inflation. The applied pressures within the SPM typically fall within the range of 100–160 kPa.

## 5. Finite Element Analysis FEA and Material Characterization

### 5.1. Characterization of Material

Soft robotics encompasses a wide range of materials employed in the manufacturing process, which contribute to the remarkable characteristics of these robots, including their flexibility, controllability, and human-safe nature. Recent advancements in soft robotics have focused on various aspects, including material selection, construction geometries, control systems, modeling techniques, and production methods [30]. The field has witnessed significant progress in developing materials specifically tailored for soft robotics applications, such as silicon rubber, TBU, and Ecoflex-30 [31]. These materials offer unique properties that distinguish them from traditional materials used in additive manufacturing, which typically involve thermoplastic polymers like PLA and ABS or thermoset polymers like TBU [19,30]. Recognizing the limitations of additive manufacturing in fabricating complex structures, a molding fabrication approach has gained popularity due to its cost-effectiveness and ability to create intricate shapes [34,35]. Silicone-based elastomers are commonly used in this approach. For our soft pneumatic muscle fabrication, silicon synthetic rubber has been selected as the primary material, known for its hyperelastic performance and high tolerance for large pressures. This material exhibits excellent mechanical properties, particularly in terms of stress–strain behavior. In order to characterize the mechanical behavior of the material, we conducted mechanical tests, specifically a uniaxial test performed under stable conditions. The uniaxial test was conducted in accordance with the ASTM No. D412 specifications, utilizing dumbbell-shaped specimens [41], as shown in Figure 3. The mold for fabricating the specimens was created using PLA material through a 3D printer, ensuring precise and consistent specimen geometries. A silicon rubber material was evaluated and tested by a uniaxial test, as illustrated in Figure 4, in which we use results from the experimental uniaxial test.

In order to comprehensively characterize the mechanical behavior of the silicon rubber material, a series of mechanical tests were conducted, including a uniaxial test performed under stable conditions. The test followed the ASTM No. D412 specifications, utilizing dumbbell-shaped specimens, as depicted in Figure 3 [41]. The mold for fabricating these specimens was created using PLA material via a 3D printer, ensuring precision and consistency in specimen geometries. Subsequently, the silicon rubber material was subjected to the uniaxial test, with the results illustrated in Figure 4.

Upon completing the experimental phase, the most suitable hyperelastic model was determined through curve fitting analysis. The Yeoh 3rd-order model was identified as the best fit for the silicon rubber material. The Yeoh model can be represented by the following equation:(1)$$\Psi =C_{10}\left(\lambda_{1}^{2}+\lambda_{2}^{2}+\lambda_{3}^{2}-3\right)+C_{20}\left(\lambda_{1}^{4}+\lambda_{2}^{4}+\lambda_{3}^{4}-3\right)+C_{30}\left(\lambda_{1}^{6}+\lambda_{2}^{6}+\lambda_{3}^{6}-3\right)$$
where Ψ is the strain energy density function; $\lambda_{1}$, $\lambda_{2}$, and $\lambda_{3}$ are the principal stretches; and $C_{10}$, $C_{20}$, and $C_{30}$ are the material constants. We then integrated the material parameters obtained from the curve-fitting process into the FEA software (ANSYS) in order to perform the finite element analysis. Table 2 displays the specific material parameters of the hyperelastic model (i.e., $C_{10}$, $C_{20}$, and $C_{30}$) for reference and further analysis.

### 5.2. Finite Element Analysis FEA

The selection of silicon synthetic rubber as the material for fabricating the soft pneumatic muscle (SPM) was based on its exceptional hyperelastic performance, high-pressure resistance, and desirable mechanical properties, particularly its stress–strain behavior. The Yeoh 3rd-order hyperelastic model was identified as the best fit for the silicon rubber material through a rigorous curve-fitting process. This model was subsequently integrated into the FEA software, ANSYS, to facilitate further analysis. The non-linear finite element analysis was carried out using ANSYS Workbench, primarily investigating various geometric aspects. To initiate the simulations, the CAD models of the soft pneumatic muscles were directly imported into the ANSYS Design Modeler, enabling the subsequent analysis of their behavior.

The simulation was designed to characterize the muscle’s response under different conditions comprehensively. This included evaluating the maximum deformation length in the x, y, and z directions and mapping the distribution of stress and strain within the structure. A pressure range from 100 to 160 kPa was applied to capture a wide range of scenarios, incrementing in steps of 10 kPa. Specifically, the pressure was selectively applied to two of the three chambers while the remaining chamber was kept at a constant pressure.

By conducting simulations for three distinct models, each featuring different pitch chamber values (30 mm, 34 mm, and 38 mm), valuable insights were gained into the performance and behavior of the SPM under varying pressure conditions. The results obtained from this extensive analysis, as visualized in Figure 5, significantly contribute to a deeper understanding of the SPM’s mechanical characteristics, paving the way for further advancements and refinements in soft robotics.

This comparison aims to analyze the influence of pitch variation on the deformation and stress behavior of the soft pneumatic muscle (SPM) when applying pressure. Specifically, the study focuses on altering the pressure in two chambers while maintaining a constant pressure in the third chamber. Throughout this comparison, the height (H = 20 mm) and thickness (T = 7 mm) of the SPM are kept consistent. The analysis reveals noteworthy findings in terms of deformation and stress. At a pitch value of P = 30 mm, the y-axis deformation (y-def) is measured to be 113 mm. When the pitch value is increased to P = 34 mm, the y-def increases to 130 mm. Further raising the pitch to P = 38 mm results in a higher y-def of 163 mm as shown in Figure 6. Consequently, deformation is significantly increased from 113 mm to 163 mm as the pitch increases.

Similarly, the stress experienced by the SPM exhibits a corresponding variation. At P = 30 mm, the stress is measured to be 0.83 MPa, whereas, at P = 38 mm, the stress escalates to 2.2 MPa. The increase in pitch leads to increased stress experienced by the SPM. It is worth noting that the deformation effect in the x and z axes is minimal in this comparison and hence does not significantly contribute to the overall analysis. The visual representation in [5] further supports this observation.

## 6. Workspace and Kinematic Model for SPM

ANNs have found extensive applications in soft robotics for controlling the behavior of soft robots, making them a powerful tool for modeling complex nonlinear systems [36,37]. ANNs enable the learning of the intricate relationship between a soft robot’s input and output parameters, enabling precise control over its behavior, motion, and stiffness, thereby facilitating shape adaptation and functional changes [36,37]. Obtaining a kinematic or dynamic model for soft robots is often challenging in model-based control systems [38,47]. To overcome this limitation, learning techniques, including neural networks, have been employed to derive accurate kinematic or dynamic models for soft robots [33,39]. These techniques give researchers a deeper understanding of the soft robot’s behavior and enhance its control mechanisms. Soft pneumatic muscles (SPMs) offer several advantages over traditional rigid actuators, such as high compliance, low weight, and the ability to generate complex multi-degree-of-freedom motions [40]. Neural networks can be effectively combined with SPMs to enhance performance and control. This is achieved by employing a neural network to model the behavior of the SPM and utilizing the model to predict the actuator’s output based on the input pressure signal [39]. This approach compensates for inherent nonlinearities and hysteresis in the SPM, improving accuracy and repeatability [39,48].

### 6.1. Workspace of Soft Pneumatic Muscles

The soft pneumatic muscles’ (SPMs) workspace was investigated using forward kinematics, established through FEA software (ANSYS) workbench simulations. This analysis provided a comprehensive understanding of the total workspace position and orientation of the SPMs in Cartesian coordinates. The workspace assessment was crucial in evaluating the range of motion and capabilities of the SPMs across a full 360-degree span. The obtained workspaces demonstrated the SPMs’ ability to perform various motions, including extension, bending, and twisting, within the Cartesian coordinate system. As depicted in Figure 7, these workspaces visually represented the SPM model’s capacity to achieve different configurations and motions throughout its surrounding area. The data used to generate these workspaces were extracted from the FEA simulations conducted using ANSYS, ensuring accuracy and reliability in assessing the SPMs’ capabilities. By examining the workspaces, valuable insights were gained regarding the SPMs’ motion capabilities and potential applications in different scenarios. The ability to perform multi-directional movements opens up possibilities for various tasks, including object manipulation, grasping, and locomotion in soft robotics applications. The characterization of the workspaces not only highlights the SPMs’ versatility but also serves as a foundation for future design and control strategies. By understanding the range of achievable motions, engineers and researchers can optimize the SPMs’ performance by tailoring their design parameters and actuation strategies accordingly.

### 6.2. Forward and Inverse Kinematic Modeling by ANN

The kinematic model of soft pneumatic muscles (SPMs) encompasses both forward and inverse kinematics. Forward kinematics calculates the X, Y, and Z coordinates of the tip center point based on a given pressure value, while inverse kinematics determines the input pressure required to achieve a desired tip position (Figure 8). In this study, we employed an artificial neural network (ANN) to model the inverse kinematics, establishing the relationship between the applied pressure in each muscle chamber and the resulting tip position.

For training the ANN, we utilized a dataset consisting of 66 samples, which included finite element analysis (FEA) results for pressures P1, P2, and P3, as well as deformation in the X, Y, and Z axes. To ensure a robust and accurate model, we adopted a Bayesian regularization (BR) approach. This type of ANN is particularly suitable for quantitative studies with smaller datasets as it can handle complex relationships without compromising power or precision. To assess the model’s performance, the dataset was divided into training, validation, and test sets, with a recommended split ratio of 70%, 15%, and 15%, respectively. The optimal ANN structure, comprising one hidden layer with ten neurons, was determined through iterative adjustments during training to minimize the mean squared error (MSE). After 300 iterations, the best-performing ANN achieved a validation MSE of 2.8655, demonstrating its accuracy in predicting pressure values for desired tip positions.

The inverse kinematics modeling employed an ANN with three inputs representing Cartesian X, Y, and Z coordinates, and three outputs representing the corresponding pressure values. Using the MATLAB neural network toolbox, the ANN was trained by iteratively adjusting the network output to optimize validation performance, as quantified by the lowest MSE. After 300 iterations, the trained ANN demonstrated an MSE of 2.8655, indicating its accuracy in predicting pressure values for desired tip positions. The training, validation, and testing process of the ANN is depicted in Figure 8.

To validate the ANN’s performance, a comparison was made between its results and finite element analysis (FEA) simulations. The observed maximum errors in the X, Y, and Z coordinates were 9.3%, 7.83%, and 8.8%, respectively, demonstrating the network’s capability to accurately predict future values. These results provide validation for the reliability and performance of the trained ANN, underscoring its effectiveness in predicting the required pressure values to achieve desired tip positions in the SPMs. Leveraging the predictive capabilities of the ANN enables the precise control and manipulation of SPMs, contributing to the advancement of soft robotics and its broad range of applications.

A comprehensive comparison was conducted between the mean absolute error (MAE) associated with two predictive models: finite element analysis (FEA) and artificial neural network (ANN). The primary objective was to predict the behavior of soft pneumatic muscles by considering key performance parameters such as force output, range of motion, and pressure response. Visual representation in the form of a bar chart was employed to illustrate the disparities in accuracy between the models, ultimately revealing the ANN model’s consistent superiority in prediction accuracy. This observation highlights the enhanced capability of the ANN model in capturing the intricate and nonlinear relationships between design parameters and muscle performance. The MAE values obtained for the ANN model were 0.2251, 0.3739, and 0.3761 for the x, y, and z directions, respectively, as indicated in the corresponding Figure 9. These significant findings offer valuable insights that support ongoing research efforts, facilitating the refinement of soft pneumatic muscle design and the exploration of its potential applications in the field of upper limb assistive devices.

## 7. Conclusions

In this study, the development and analysis of three soft pneumatic muscles (SPMs) were conducted in order to explore their potential use in human assistive devices for individuals with disabilities. The SPMs’ geometric characteristics were meticulously designed and presented in Table 1. By employing simulation techniques using ANSYS, the final positions of the SPMs were determined. The simulations encompassed various pressure conditions ranging from 100 to 160 kPa, wherein two muscles underwent pressure changes while the remaining muscle remained constant. Under constant chamber height (H) and chamber thickness (T) conditions, the finite element analysis (FEA) outcomes provided valuable insights into the deformation and stress behaviors exhibited by the SPMs. Notably, an examination of the deformation and stress characteristics revealed the influential role of the chamber pitch. Increasing the chamber pitch from P = 30 mm to P = 38 mm resulted in a proportional rise in deformation from 113 mm to 163 mm, accompanied by an increase in stress from 0.83 MPa to 2.2 MPa. This comparative analysis emphasized the critical importance of meticulous design considerations. Furthermore, an inverse kinematic model was established, utilizing the tip positions (x, y, and z) of the SPMs as inputs to predict the corresponding pressure values at desired positions. To achieve this, an ANN was implemented, yielding promising results and showcasing the trained ANN’s excellent performance. The accuracy of the ANN model was supported by a mean squared error (MSE) of 2.8655, while the training, validation, and testing residuals further reinforced its reliability, measuring 99.58%, 99.89%, and 99.79%, respectively. Notably, the validation simulation and neural network results exhibited maximum errors of 9.3% in the x direction, 7.83% in the y direction, and 8.8% in the z direction. To further advance this research, additional avenues for future work can be explored. One such direction involves the application of practical work, entailing the execution of experimental tests on physical prototypes. These tests would provide a more accurate understanding of the SPMs’ behavior within real-world scenarios. Another potential area of investigation lies in incorporating PID control schemes to regulate and control the SPMs. This integration could potentially enhance their precision and stability during operation. Moreover, the utilization of optimization modeling techniques, particularly through the application of deep learning methods, shows promise in improving the prediction capabilities of the inverse kinematic model.

## Figures and Tables

**Figure 1 micromachines-14-01431-f001:**
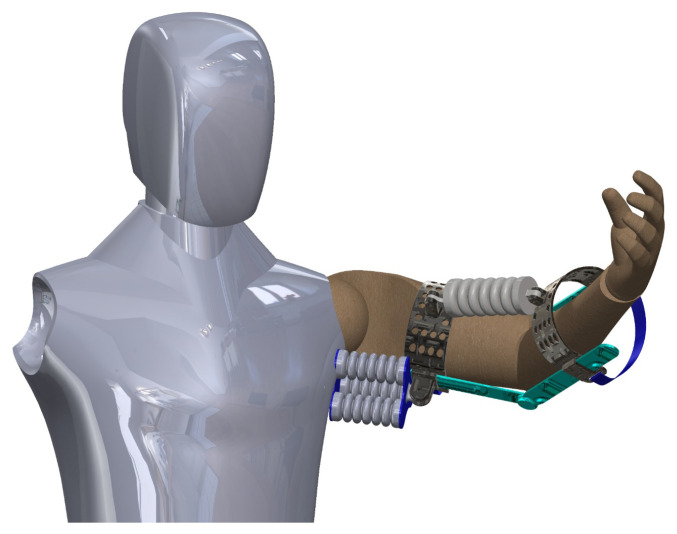
Schematic representation of an upper limb soft wearable exoskeleton, highlighting the integration of three parallel soft pneumatic muscles for efficient shoulder joint actuation and a single-chamber SPMfor elbow joint rotation.

**Figure 2 micromachines-14-01431-f002:**
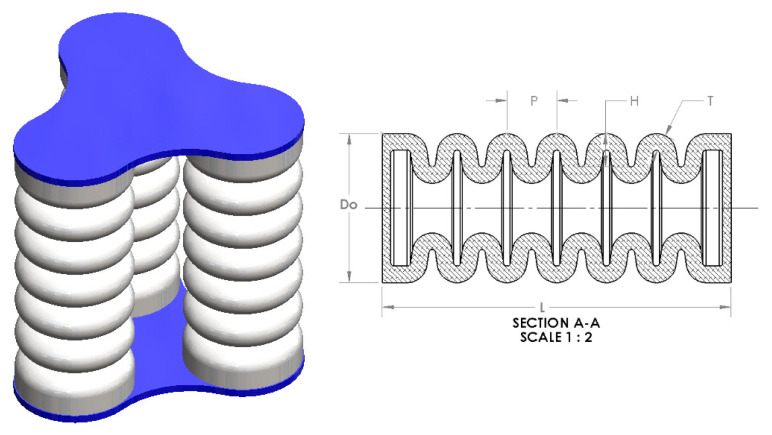
3D model and parameters of a soft pneumatic muscle. The figure presents a visual representation of the soft pneumatic muscle, showcasing its design and highlighting important parameters.

**Figure 3 micromachines-14-01431-f003:**
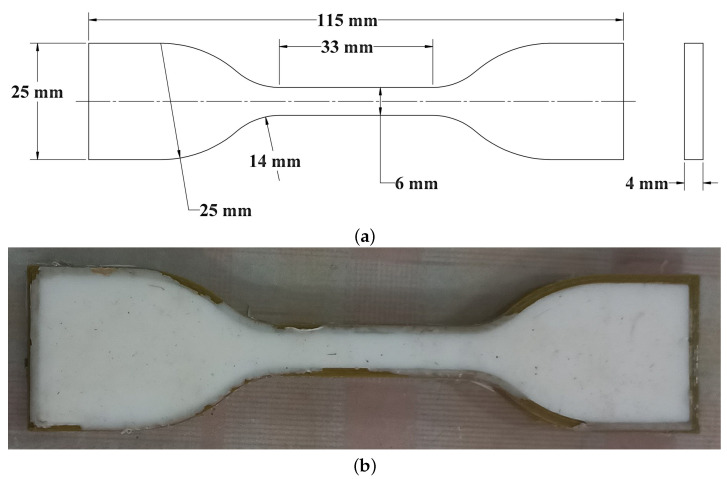
Geometrical dimensions and manufacturing of ASTM D412 specimen of silicone rubber material. (**a**) Detailed drawing of standard specimen ASTM D412 [41]. (**b**) Fabricated ASTM D412 specimen of silicone rubber after curing.

**Figure 4 micromachines-14-01431-f004:**
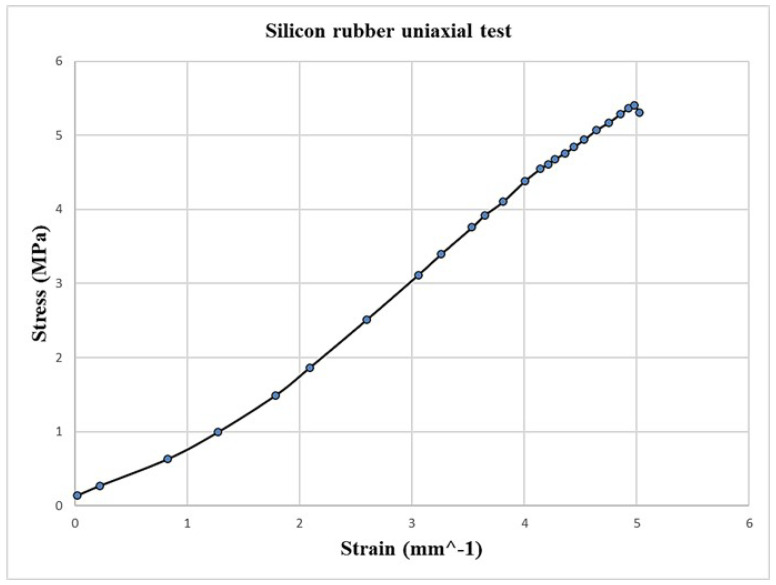
Experimental results: Stress—Strain curve for the tested silicone rubber material.

**Figure 5 micromachines-14-01431-f005:**
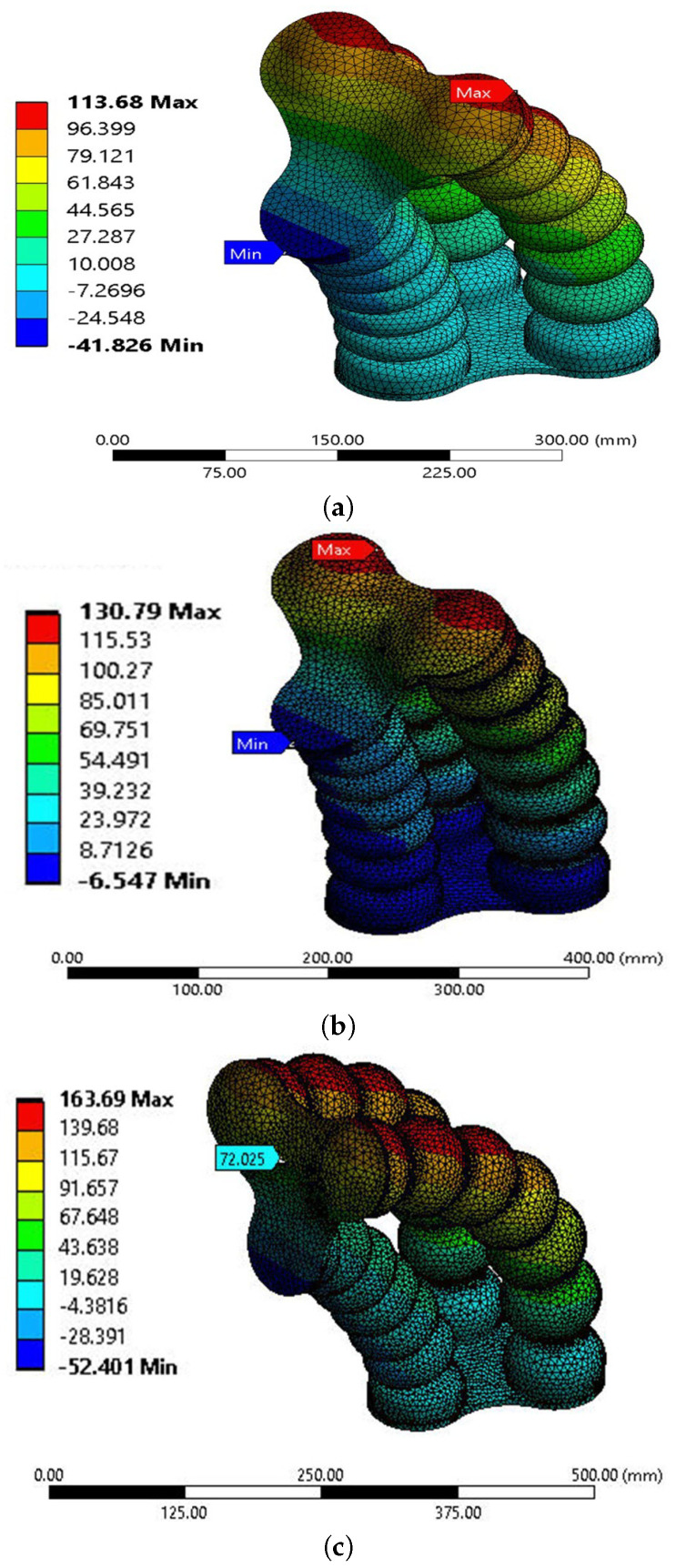
Demonstrate FEA results of three tests for in Y-Deformation at different chamber pitches at max pressure value. (**a**) Demonstrate Y-Deformation for P = 30 mm, (**b**) Y-Deformation for P = 34 mm, (**c**) Y-Deformation for P = 38 mm.

**Figure 6 micromachines-14-01431-f006:**
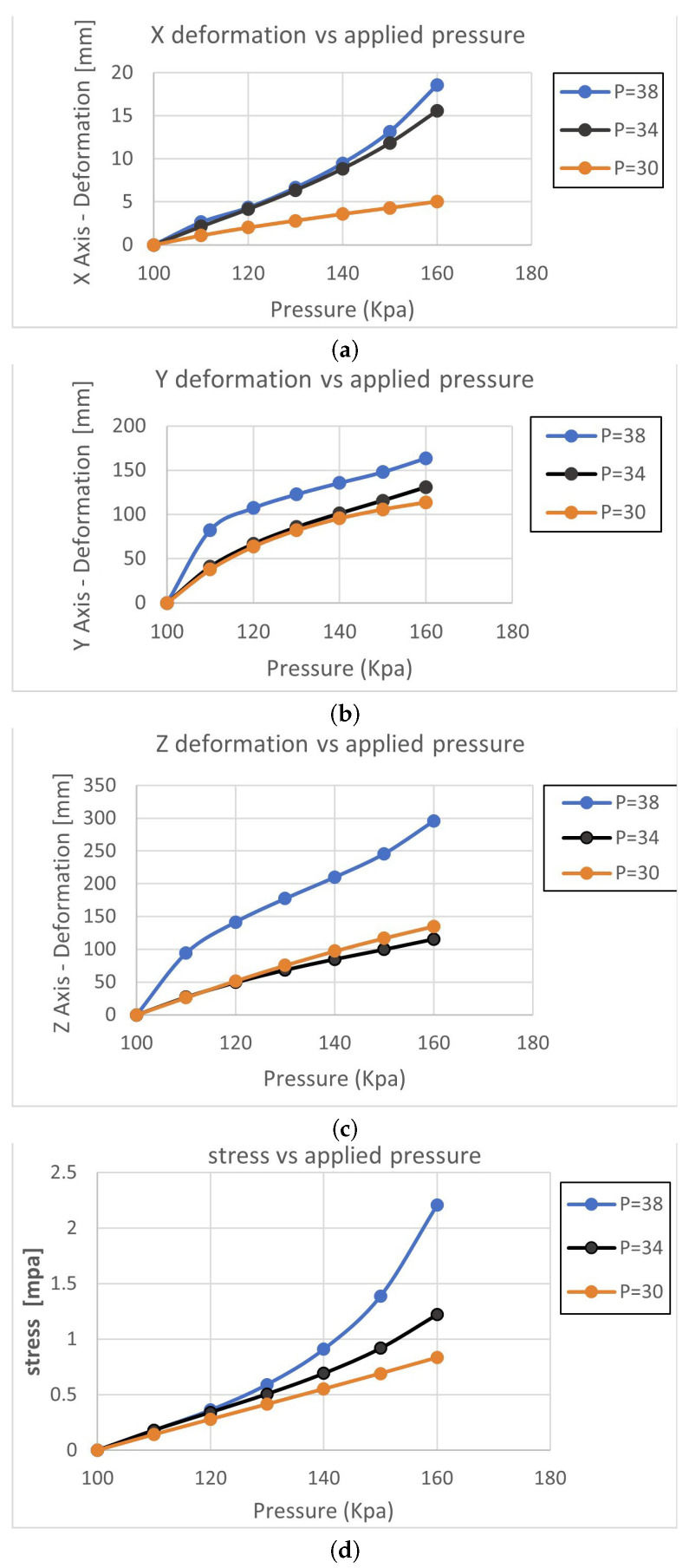
Demonstration of the FEA Results of three tests at various applied pressures. (**a**) demonstration of x-axis deformation vs. applied pressure; (**b**) demonstration of y-axis deformation vs. applied pressure; (**c**) demonstration of z-axis deformation vs. applied pressure; (**d**) demonstration of the stress vs. applied pressure.

**Figure 7 micromachines-14-01431-f007:**
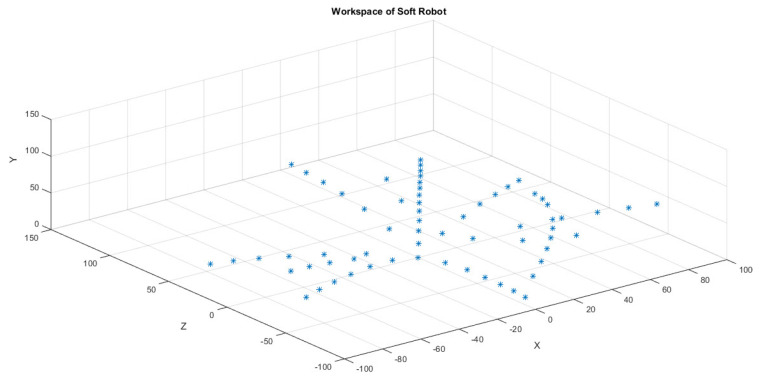
The workspace of SPM.

**Figure 8 micromachines-14-01431-f008:**
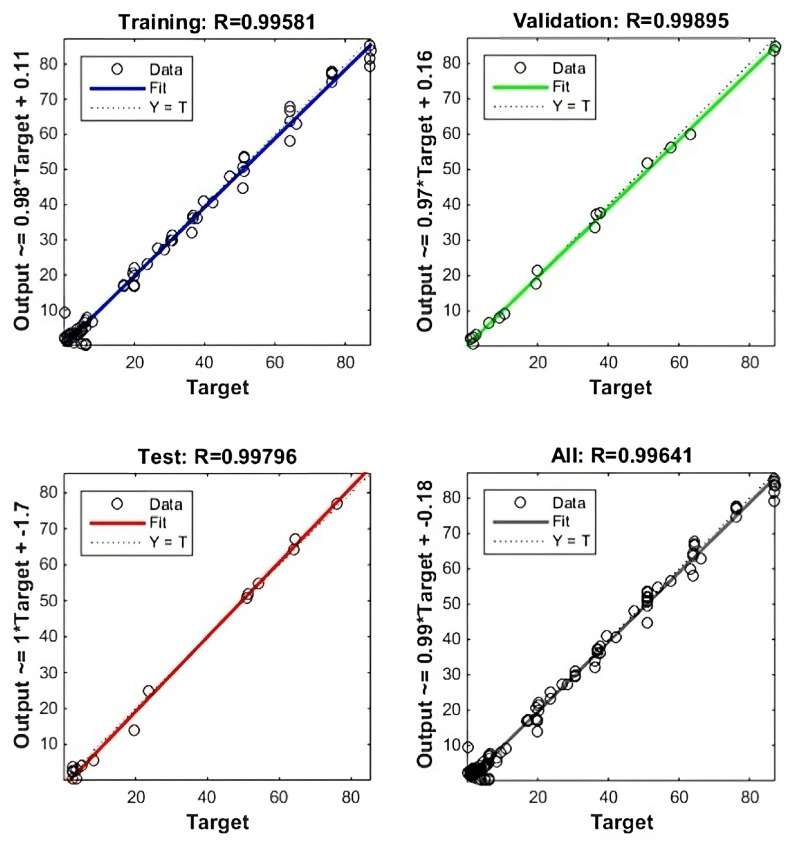
The training, validation, testing, and all residuals values of ANN.

**Figure 9 micromachines-14-01431-f009:**
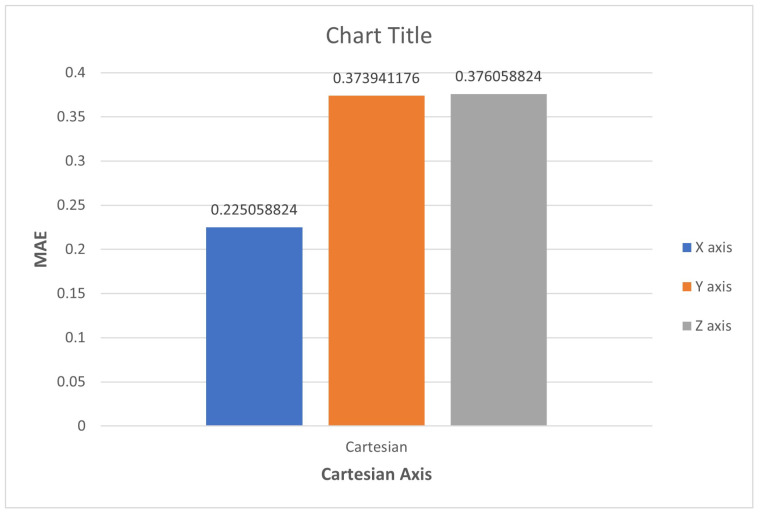
Bar chart illustrating the mean absolute error (MAE) values associated with the finite element analysis (FEA) and artificial neural network (ANN) models for predicting the behavior of soft pneumatic muscles.

**Table 1 micromachines-14-01431-t001:** The geometrical parameters of SPM.

Parameters	L	$D_{o}$	T	H	P
Values	210 mm	90 mm	7 mm	20 mm	30 to 38 mm

**Table 2 micromachines-14-01431-t002:** The parameters of the Yeoh 3rd order model of silicon rubber material.

Description	Value
Silicon Rubber Density	1080 kg/m^3^
$C_{10}$	0.196259775 MPa
$C_{20}$	0.009010431 MPa
$C_{30}$	−0.000105654 MPa

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
