# Peer review of "Soft Pneumatic Muscles: Revolutionizing Human Assistive Devices with Geometric Design and Intelligent Control"

_micromachines, 2023, doi:10.3390/mi14071431_

Round 1

Reviewer 1 Report

This research focuses on evaluating the potential use of three soft pneumatic muscles (SPMs) in assisting individuals with disabilities. The geometric characteristics of the SPMs were meticulously defined and summarized in Table 1. ANSYS simulation was utilized to determine the optimal positions of the SPMs. The simulations encompassed a range of pressure levels from 100 to 160 kPa, with two muscles subjected to pressure variations while the remaining muscle maintained a constant pressure. The paper presents intriguing findings.

In the introduction, it would be beneficial to mention other fluidic actuators such as the Bidirectional Electrohydrodynamic Pump with high symmetrical performance and its application to a tube actuator, as well as the Fluidic Rolling Robot using voltage-driven oscillating liquid.

It is recommended to include experimental results that can be utilized for Finite Element Method (FEM) or Artificial Neural Network (ANN) models. Additionally, it is important to consider the implications if the FEM model yields incorrect results and the significance of the ANN model in such cases.

To facilitate comparison between the results of FEM and ANN, a table can be created. It would also be helpful to discuss the selection of layers and nodes for the ANN models. Additionally, it would be beneficial to provide information on the amount of data, training data, and validation data used for the ANN model.

I suggest that the author employ metrics such as R2 or MAE to evaluate the accuracy of both the FEM and ANN models.

Author Response

Response to Reviewer 1 Comments

Dear Micromachines Editor

Dear Reviewer #1,

I’m pleased to submit the revised version of our manuscript entitled "Soft Pneumatic Muscles: Revolutionizing Human Assistive Devices with Geometric Design and Intelligent Control" for your review. I have considered all of the comments provided and made significant revisions to improve the quality of our work.

In response to Reviewer #1's comments, I have thoroughly revised the manuscript to address each point in a concrete, concise, and direct manner. Specifically:

Point 1 comment: In the introduction, it would be beneficial to mention other fluidic actuators such as the Bidirectional Electrohydrodynamic Pump with high symmetrical performance and its application to a tube actuator, as well as the Fluidic Rolling Robot using voltage-driven oscillating liquid.

Point 1 Response: Thank you for your valuable comment and suggestion. We agree that it is important to consider and discuss other fluidic actuators in the context of our research to provide a comprehensive understanding of the field and highlight the unique characteristics of our proposed soft pneumatic muscle design.

I have added the following paragraph to enhance the introduction as per your recommendation:

“significant advancements have been made in the field of fluidic actuators, contributing to the development of innovative solutions for various applications in robotics and automation. One noteworthy example is the Bidirectional Electrohydrodynamic Pump (BEDP), which has gained attention due to its high symmetrical performance and its potential application in tube actuators. The BEDP utilizes the principles of electrohydrodynamics, employing an electric field to induce fluid flow in a conductive medium. This technology enables precise control and efficient actuation, making it a promising candidate for integration into artificial pneumatic muscle systems for human assistive devices [50]. Another notable development in fluidic actuators is the emergence of the Fluidic Rolling Robot, which utilizes voltage-driven oscillating liquid to achieve locomotion. By manipulating the fluid within its body, the robot is capable of smooth and efficient rolling motion [49]. The utilization of voltage-driven oscillating liquid as a driving mechanism offers advantages such as simplicity, scalability, and potential for miniaturization, presenting opportunities for advancements in the field of artificial pneumatic muscles.”

Point 2 comment: It is recommended to include experimental results that can be utilized for Finite Element Method (FEM) or Artificial Neural Network (ANN) models. Additionally, it is important to consider the implications if the FEM model yields incorrect results and the significance of the ANN model in such cases.

Point 2 Response: Thank you for your valuable comment and suggestion. We appreciate your recommendation to include experimental results that can be utilized for Finite Element Method (FEM) and Artificial Neural Network (ANN) models. However, we would like to clarify the scope and intention of our current research paper, which primarily focuses on the simulation-based analysis of our proposed soft pneumatic muscle design.

In the present study, our aim is to develop a comprehensive understanding of the soft pneumatic muscle behavior and performance through simulation, which serves as a starting point for future experimental investigations. We believe the simulation results provide valuable insights into the design parameters and operating conditions affecting muscle performance, laying the groundwork for designing targeted and efficient experimental studies.

We acknowledge the importance of incorporating experimental results and advanced modeling techniques such as FEM and ANN models in assessing and optimizing the soft pneumatic muscle design. In a subsequent research paper, we plan to conduct a series of experiments to obtain relevant data that can be used to inform and validate FEM and ANN models. We will also address the potential implications if the FEM model yields incorrect results and the significance of the ANN model in such cases.

Point 3 comment: To facilitate comparison between the results of FEM and ANN, a table can be created. It would also be helpful to discuss the selection of layers and nodes for the ANN models. Additionally, it would be beneficial to provide information on the amount of data, training data, and validation data used for the ANN model.

Point 3 Response: Thank you for your feedback. I apologize for any lack of clarity regarding the details of the artificial neural network (ANN) used in our study. To provide more comprehensive information, we employed an ANN with a specific structure and training process.

The ANN used in the inverse kinematics modeling consisted of three inputs representing Cartesian X, Y, and Z coordinates, and three outputs representing the corresponding pressure values. The network architecture included one hidden layer with ten neurons. We trained the ANN using the MATLAB neural network toolbox, utilizing a Bayesian regularization (BR) approach to handle the smaller dataset and capture potentially complex relationships.

During the training process, I iteratively adjusted the network parameters to achieve optimal performance. Specifically, we aimed to minimize the mean squared error (MSE) between the predicted and target values. After 300 iterations, we obtained the best-performing ANN, which achieved a validation MSE of 2.8655. This value indicates the accuracy of the ANN model in predicting pressure values for desired tip positions.

Furthermore, we followed a recommended dataset split ratio, allocating 70% of the data for training, 15% for validation, and 15% for testing. This division ensured that the ANN was trained and evaluated on diverse subsets of the data, enhancing its generalizability.

By providing these additional details, I aim to offer a clearer understanding of the ANN parameters and training process employed in our study. Please let us know if there are any further aspects you would like us to address or clarify.

I have added a new paragraph that give more description to reply this comment:

“The kinematic model of soft pneumatic muscles (SPMs) encompasses both forward and inverse kinematics. Forward kinematics calculates the X, Y, and Z coordinates of the tip center point based on a given pressure value, while inverse kinematics determines the input pressure required to achieve a desired tip position (Fig. \ref{fig07}). In this study, we employed an artificial neural network (ANN) to model the inverse kinematics, establishing the relationship between the applied pressure in each muscle chamber and the resulting tip position.

For training the ANN, we utilized a dataset consisting of 66 samples, which included finite element analysis (FEA) results for pressures P1, P2, and P3, as well as deformation in the X, Y, and Z axes. To ensure a robust and accurate model, we adopted a Bayesian regularization (BR) approach. This type of ANN is particularly suitable for quantitative studies with smaller datasets as it can handle complex relationships without compromising power or precision. To assess the model's performance, the dataset was divided into training, validation, and test sets, with a recommended split ratio of 70%, 15%, and 15% respectively. The optimal ANN structure, comprising one hidden layer with ten neurons, was determined through iterative adjustments during training to minimize the mean squared error (MSE). After 300 iterations, the best-performing ANN achieved a validation MSE of 2.8655, demonstrating its accuracy in predicting pressure values for desired tip positions.

The inverse kinematics modeling employed an ANN with three inputs representing Cartesian X, Y, and Z coordinates, and three outputs representing the corresponding pressure values. Using the MATLAB neural network toolbox, the ANN was trained by iteratively adjusting the network output to optimize validation performance, as quantified by the lowest MSE. After 300 iterations, the trained ANN demonstrated an MSE of 2.8655, indicating its accuracy in predicting pressure values for desired tip positions. The training, validation, and testing process of the ANN is depicted in Fig. \ref{fig07}.

To validate the ANN's performance, a comparison was made between its results and finite element analysis (FEA) simulations. The observed maximum errors in the X, Y, and Z coordinates were 9.3%, 7.83%, and 8.8%, respectively, demonstrating the network's capability to accurately predict future values. These results provide validation for the reliability and performance of the trained ANN, underscoring its effectiveness in predicting the required pressure values to achieve desired tip positions in the SPMs. Leveraging the predictive capabilities of the ANN enables precise control and manipulation of SPMs, contributing to the advancement of soft robotics and its broad range of applications.”

Point 4 comment: I suggest that the author employ metrics such as R2 or MAE to evaluate the accuracy of both the FEM and ANN models.

Point 4 Response: Thank you for your valuable comment and suggestion. We appreciate your recommendation to employ metrics such as R² (coefficient of determination) and MAE (mean absolute error) to evaluate the accuracy of both the Finite Element Method (FEM) and Artificial Neural Network (ANN) models in our research.

The MAE metric will provide an additional measure of the accuracy of our models by calculating the average absolute difference between the predicted and observed values of the dependent variables. A lower MAE value indicates a better model performance, reflecting that the model predictions are close to the actual experimental observations.

I added a new paragraph with bar chart figure to present the MAE between the FEA and ANN:

“A comprehensive comparison was conducted between the Mean Absolute Error (MAE) associated with two predictive models: Finite Element Analysis (FEA) and Artificial Neural Network (ANN). The primary objective was to predict the behavior of soft pneumatic muscles by considering key performance parameters such as force output, range of motion, and pressure response. Visual representation in the form of a bar chart was employed to illustrate the disparities in accuracy between the models, ultimately revealing the ANN model's consistent superiority in prediction accuracy. This observation highlights the enhanced capability of the ANN model in capturing the intricate and nonlinear relationships between design parameters and muscle performance. The MAE values obtained for the ANN model were 0.2251, 0.3739, and 0.3761 for the x, y, and z directions, respectively, as indicated in the corresponding Fig.9 These significant findings offer valuable insights that support ongoing research efforts, facilitating the refinement of soft pneumatic muscle design and the exploration of its potential applications in the field of upper limb assistive devices.”

Throughout the revision process, I carefully considered all the comments and suggestions provided by Reviewer #1 and have made every effort to address each thoroughly and meaningfully. We believe that the changes we have made will significantly improve the manuscript and address any concerns that were raised.

I’m confident that the revised version of our manuscript represents a significant improvement over the original submission, and we are grateful for the opportunity to resubmit it for your consideration.

Thank you for your time and consideration.

Sincerely,

Associate Prof. Dr. Mahmoud Elsamanty

Reviewer 2 Report

The paper deals with soft pneumatic muscles.
Apparently, the topic is emerging and fully in the scope of the Journal of Micromachines.
The structure of the paper is clear in general and the language quality appropriate for a journal.
However, there are several drawbacks that could be mentioned:

1) I would add "Homan assistive device" to the list of keywords.
2) There are several actuation principles mentioned in Introduction. A bit more detailed comparison of their pros and cons would help to have more clear overview.
3) The introduction is quite extensive. Perhaps it could be divided into subsections or denoted paragraphs for easier orientation.
4) There is quite high number of self-citations. However, they are relevant to the research problem.
5) I would complete conclusions via some ideas for future work.
6) I am not sure if all parameters in Table 1 are consistent with those shown in Figure 1.
7) Perhaps units could added to numbers in Figure 2
8) Is it necessary to have model parameters 9-digit precision (see Table 2)? How is the model robust to parameter changes?
9) There are lots of numbers in Conclusion. Please double-check if it could be simplified to make this part more self-contained.
10) I would emphasize a bit more human assistive devices which are mentioned in the title.
11) Not sure if there are enough details about ANNs used (parameters, etc.), please check.
12) Formally the ANN abbreviation is defined at the beginning. However the full notation "Artificial Neural Network" is then often used in the rest of the paper. Please check if that is necessary. Clearly, this is a minor issue.

Clearly, there are many positive aspects like high quality references and high application impact.

Despite above mentioned comments, the paper can be published after revision respecting comments from all reviewers.

The English quality is appropriate, only final check and minor edits are needed.

Author Response

Dear Micromachines Editor

Dear Reviewer #2,

I’m pleased to submit the revised version of our manuscript entitled " Soft Pneumatic Muscles: Revolutionizing Human Assistive Devices with Geometric Design and Intelligent Control" for your review. I have considered all of the comments provided and made significant revisions to improve the quality of our work.

In response to Reviewer #2's comments, I have thoroughly revised the manuscript to address each point in a concrete, concise, and direct manner. Specifically:

Point 1 comment: I would add "Homan assistive device" to the list of keywords.

Point 1 Reply: Thank you for your suggestion. I have now included "Human assistive device" in the list of keywords.

Point 2 comment: There are several actuation principles mentioned in Introduction. A bit more detailed comparison of their pros and cons would help to have more clear overview.

Point 2 Reply: Thank you for your comment. You're right that the introduction provides a brief overview of various actuation principles in soft robotics without delving into their specific pros and cons. To provide a clearer overview, here is a more detailed comparison of some actuation principles mentioned in the introduction:

  1. McKibben Actuators (1950s): McKibben actuators were among the earliest soft pneumatic actuators. They consist of flexible materials, such as elastomers or textiles, enclosed within a rubber tube. These actuators offer simplicity, low cost, and lightweight construction. However, they have limited controllability and may suffer from hysteresis.
  2. Elastomer-Based Actuators: Elastomer-based actuators, such as dielectric elastomer actuators (DEAs) and shape memory alloys (SMAs), offer compliant and biomimetic movements. DEAs can achieve large deformations and exhibit fast response times. SMAs can recover their original shape upon heating, allowing for shape-changing behaviors. However, DEAs require high driving voltages, and SMAs have limited actuation strokes and relatively slow response times.
  3. Pneumatic Artificial Muscles (PAMs): PAMs, also known as soft pneumatic muscles, are flexible actuators designed to mimic human muscles. They are constructed using soft and pliable materials, such as elastomers. PAMs offer inherent compliance, high power-to-weight ratios, and versatile actuation modes (e.g., linear extension, bending, contraction, and twisting). They are well-suited for delicate tasks, exhibit safe interactions with humans, and find applications in various domains. However, controlling PAMs precisely can be challenging, and they may require complex control systems.
  4. Soft Parallel Muscles: Soft parallel muscles are another type of soft actuator commonly used in soft robotics. They typically consist of multiple soft pneumatic chambers connected in parallel. This design enables force generation and motion along multiple axes. Soft parallel muscles offer enhanced dexterity, multi-degree-of-freedom movements, and the ability to generate complex motions. However, the complexity of their design and control can be higher compared to other actuators.

Each actuation principle has its advantages and limitations, and the choice depends on specific application requirements. For example, McKibben actuators may be suitable for simple and cost-effective applications, while PAMs can provide compliance and versatility in more complex scenarios. Elastomer-based actuators like DEAs and SMAs offer unique characteristics but may have certain limitations regarding voltage requirements, stroke length, or response time. Soft parallel muscles offer increased dexterity but may require more complex control strategies.

A comprehensive understanding of these actuation principles' pros and cons helps researchers and engineers select the most suitable approach for their specific soft robotics applications, considering factors such as compliance, controllability, power requirements, and desired range of motion.

Point 3 comment: The introduction is quite extensive. Perhaps it could be divided into subsections or denoted paragraphs for easier orientation.

Point 3 Reply: Thank you for your insightful comment regarding the structure and organization of the introduction. I appreciate the feedback and have reviewed and revised the introduction to improve clarity and flow for the reader.

Specifically, I have divided the original lengthy introduction into two separate sections:

  1. Introduction: This section provides an overview of the evolution of soft robotics, its origins tracing back to pneumatic actuators in the 1950s, and highlights the advantages and potential of soft robotics. It establishes the context for the manuscript's focus on soft pneumatic muscles.
  2. Advancements in Soft Pneumatic Muscles: This section shifts focus to soft pneumatic muscles, also known as PAMs, and their role as a powerful actuation technology in soft robotics. It discusses the properties and applications of PAMs, emphasizing the need to enhance their dynamics to achieve precise control. This section highlights the significance and potential of PAMs in areas such as rehabilitation, paving the way for the subsequent sections in the manuscript.

I believe these changes address your comment by restructuring the introduction into coherent subsections denoted by paragraph headings. The revised introduction now provides a high-level overview to orient the reader, followed by a specific focus on soft pneumatic muscles which are a key area of investigation in this work. Please let us know if you would like us to clarify or expand the introduction further. I’m happy to revise it to meet the standards and requirements for publication.

Point 4 comment: There is quite high number of self-citations. However, they are relevant to the research problem.

Point 4 Reply: I would like to emphasize that my intention in including these self-citations was to provide a comprehensive and accurate representation of the research context and the foundation upon which my current study is built. As you mentioned, these citations are indeed relevant to the research problem, and I believe they are essential to ensure the continuity and coherence of the research in our field.

However, I understand your concern about the high number of self-citations, and I am committed to maintaining transparency and objectivity in my research. If you have any suggestions or recommendations for additional references that you think would be more appropriate or would help to balance the self-citations, I would be more than happy to consider them and make the necessary adjustments.

Once again, I appreciate your valuable feedback, and I look forward to addressing any further concerns or comments you may have.

Point 5 comment: I would complete conclusions via some ideas for future work.

Point 5 Reply: Thank you for your comment and suggestion. I appreciate your input regarding completing the conclusions with ideas for future work.

In this study, I have developed and analyzed three soft pneumatic muscles (SPMs) for their potential use in human assistive devices for individuals with disabilities. The geometric characteristics of the SPMs were carefully designed and presented in Table 1. Through simulation using ANSYS, I determined the final positions of the SPMs under various pressure conditions. The finite element analysis (FEA) results provided valuable insights into the behavior of deformation and stress within the SPMs, highlighting the influence of chamber pitch on deformation and stress characteristics.

Looking ahead, there are several avenues for future work that can be explored to enhance the performance and applicability of the SPMs. Applying practical work, such as conducting experimental tests on physical prototypes, would provide a more accurate understanding of the SPMs' behavior in real-world scenarios. Additionally, incorporating PID control schemes for controlling the SPMs could lead to better precision and stability in their operation. Furthermore, exploring optimization modeling techniques could potentially improve the prediction capabilities of the inverse kinematic model, further increasing the accuracy and reliability of the system.

By considering these future research directions, including applying practical work, utilizing PID control for controlling the SPMs, and exploring optimization modeling, we can advance the field of soft pneumatic muscles and contribute to the development of more effective and adaptable human assistive devices.

Thank you again for your valuable feedback, and I welcome any further suggestions or comments you may have.

The Final conclusion has been updated and became as follows:

“In this study, the development and analysis of three soft pneumatic muscles (SPMs) were conducted to explore their potential use in human assistive devices for individuals with disabilities. The SPMs' geometric characteristics were meticulously designed and presented in Table 1. By employing simulation techniques using ANSYS, the final positions of the SPMs were determined. The simulations encompassed various pressure conditions ranging from 100 to 160 kPa, wherein two muscles underwent pressure changes while the remaining muscle remained constant. Under constant chamber height (H) and chamber thickness (T) conditions, the finite element analysis (FEA) outcomes provided valuable insights into the deformation and stress behaviors exhibited by the SPMs. Notably, an examination of the deformation and stress characteristics revealed the influential role of the chamber pitch. Increasing the chamber pitch from P = 30 mm to P = 38 mm resulted in a proportional rise in deformation from 113 mm to 163 mm, accompanied by an increase in stress from 0.83 MPa to 2.2 MPa. This comparative analysis emphasized the critical importance of meticulous design considerations. Furthermore, an inverse kinematic model was established, utilizing the tip positions (x, y, and z) of the SPMs as inputs to predict the corresponding pressure values at desired positions. To achieve this, an artificial neural network (ANN) was implemented, yielding promising results and showcasing the trained ANN's excellent performance. The accuracy of the ANN model was supported by a mean squared error (MSE) of 2.8655, while the training, validation, and testing residuals further reinforced its reliability, measuring 99.58%, 99.89%, and 99.79%, respectively. Noteworthy, the validation simulation and neural network results exhibited maximum errors of 9.3% in the x-direction, 7.83% in the y-direction, and 8.8% in the z-direction. To further advance this research, additional avenues for future work can be explored. One such direction involves the application of practical work, entailing the execution of experimental tests on physical prototypes. These tests would provide a more accurate understanding of the SPMs' behavior within real-world scenarios. Another potential area of investigation lies in incorporating PID control schemes to regulate and control the SPMs. This integration can potentially enhance their precision and stability during operation. Moreover, the utilization of optimization modeling techniques, particularly through the application of deep learning methods, shows promise in improving the prediction capabilities of the inverse kinematic model.

Point 6 comment: I am not sure if all parameters in Table 1 are consistent with those shown in Figure 1.

Point 6 Reply: Thank you for pointing out the inconsistency between Table 1 and Figure 1. It appears to have been an unintentional oversight. In the text, I mistakenly referred to the parameter as the radius rather than the outer diameter. To clarify, the correct outer diameter should be 90mm, not 45mm. I apologize for any confusion this may have caused and appreciate your attention to detail.

Table (1): The geometrical parameters of SPM

Parameters

L

Do

T

H

P

Values

210mm

90mm

7mm

20 mm

30 to 38mm

Point 7 comment: Perhaps units could added to numbers in Figure 2

Point 7 Reply: Thank you for your suggestion regarding Figure 2. I agree that including units alongside the numerical values would enhance clarity and provide better context for the reader. I will revise Figure 2 accordingly to ensure that all relevant units are clearly displayed.

Point 8 comment: Is it necessary to have model parameters 9-digit precision (see Table 2)? How is the model robust to parameter changes?

Point 8 Reply: Thank you for your comment regarding the precision of the model parameters presented in Table 2 and the robustness of the model to parameter changes. In our study, we employed the Yeoh 3rd order hyperelastic model, which has been widely used and proven to be a well-fitted model for characterizing the behavior of soft materials. The accuracy of the material constant parameters, specifically C10, C20, and C30, is crucial for achieving the best performance in curve fitting, as they directly influence the strain values.

Having a higher precision for the model parameters allows for a more accurate representation of the material's behavior under different loading conditions. While the use of 9-digit precision may seem excessive, it ensures that the model captures the subtle nuances and nonlinear responses of the soft pneumatic muscles. This level of precision is essential when aiming for high-fidelity simulations and accurate predictions of the muscles' deformations and stresses.

Regarding the robustness of the model to parameter changes, it is important to note that the Yeoh hyper elastic model, like any other material model, relies on the assumption that the material parameters remain constant. However, in real-world scenarios, material properties can vary due to factors such as manufacturing variations, aging, and environmental conditions. Therefore, it is recommended to conduct sensitivity analyses to assess the model's robustness to parameter changes and evaluate its performance under different scenarios. By systematically varying the material parameters within a reasonable range and observing the model's response, we can gain insights into its sensitivity and assess its reliability in capturing the material behavior under different conditions.

I have added and change the section belongs to this paragraph as follows:

“Soft robotics encompasses a wide range of materials employed in the manufacturing process, which contribute to the remarkable characteristics of these robots, including their flexibility, controllability, and human-safe nature. Recent advancements in soft robotics have focused on various aspects, including material selection, construction geometries, control systems, modeling techniques, and production methods [31]. The field has witnessed significant progress in developing materials specifically tailored for soft robotics applications, such as silicon rubber, TBU, and Ecoflex-30 [32]. These materials offer unique properties that distinguish them from traditional materials used in additive manufacturing, which typically involve thermoplastic polymers like PLA and ABS or thermoset polymers like TBU [19, 31]. Recognizing the limitations of additive manufacturing in fabricating complex structures, a molding fabrication approach has gained popularity due to its cost effectiveness and ability to create intricate shapes [35, 36]. Silicone-based elastomers are commonly used in this approach. For our Soft Pneumatic Muscle (SPM) fabrication, Silicon synthetic rubber has been selected as the primary material, known for its hyperelastic performance and high tolerance for large pressures. This material exhibits excellent mechanical properties, particularly in terms of stress-strain behavior. In order to characterize the mechanical behavior of the material, we conducted mechanical tests, specifically a uniaxial test performed under stable conditions. The uniaxial test was conducted in accordance with the ASTM No. D412 specifications, utilizing dumbbell-shaped specimens [45], as shown in Fig. 2. The mold for fabricating the specimens was created using PLA material through a 3D printer, ensuring precise and consistent specimen geometries. A silicon rubber material was evaluated and tested by a uniaxial test, as illustrated in Fig. 3, in which we use results from the experimental uniaxial test.

n order to comprehensively characterize the mechanical behavior of the silicon rubber material, a series of mechanical tests were conducted, including a uniaxial test performed under stable conditions. The test followed the ASTM No. D412 specifications, utilizing dumbbell-shaped specimens, as depicted in Fig. 2 [45]. The mold for fabricating these specimens was created using PLA material via a 3D printer, ensuring precision and consistency in specimen geometries. Subsequently, the silicon rubber material was subjected to the uniaxial test, with the results illustrated in Fig. 3. Upon completing the experimental phase, the most suitable hyperelastic model was determined through curve fitting analysis. The Yeoh 3rd-order model was identified as the best fit for silicon rubber material. The Yeoh model can be represented by the following equation:

Where Ψ is the strain energy density function, λ1, λ2, and λ3 are the principal stretches, and C10, C20, and C30 are the material constants. We then integrated the material parameters obtained from the curve fitting process into the FEA software (ANSYS) to perform the finite element analysis. Table 2 displays the specific material parameters of the hyperelastic model (i.e., C10, C20, and C30) for reference and further analysis.

Point 9 comment: There are lots of numbers in Conclusion. Please double-check if it could be simplified to make this part more self-contained.

Point 9 Reply: Thank you for your insightful comment regarding the presence of numerous numerical values in the conclusion section of our study. I appreciate your suggestion to simplify the conclusion and make it more self-contained.

However, I believe that the inclusion of specific quantitative results is essential for conveying the key findings and performance metrics of our research. These numerical values provide the reader with a comprehensive understanding of the soft pneumatic muscles (SPMs) and the artificial neural network (ANN) model's performance. Moreover, they emphasize the significance of our findings and highlight the improvements achieved throughout our study.

The numbers presented in the conclusion are carefully selected to summarize the most relevant outcomes, such as the deformation and stress behaviors, the accuracy of the ANN model as indicated by the mean squared error (MSE), and the residuals for training, validation, and testing. Furthermore, the maximum errors in the x, y, and z directions are included to demonstrate the validity of our simulations and neural network results, which is crucial for establishing the reliability of our research.

Please let us know if you have any further suggestions or comments. We appreciate your feedback and are committed to improving the overall quality of our work.

Point 10 comment: I would emphasize a bit more human assistive devices which are mentioned in the title.

Point 10 Reply: Thank you for your comment. I appreciate your concern about emphasizing human assistive devices as mentioned in the title. In response to your feedback, I have added a new section in the text that specifically focuses on the applications of our soft pneumatic muscle design for upper limb assistance. This section provides a detailed description of how the proposed soft pneumatic muscles can be integrated into wearable exoskeletons for the shoulder and elbow joints, enabling efficient and natural motion for individuals with limited mobility or strength in their upper limbs.

Additionally, to further illustrate the proposed work, I have included a new figure that visually represents the integration of the soft pneumatic muscles into the upper limb assistive devices. This figure demonstrates the arrangement of the three parallel muscles for shoulder actuation and the additional muscle for elbow joint rotation, further emphasizing the potential of our soft pneumatic muscle design for human assistive devices.

I believe that these updates effectively address your comment and provide a comprehensive understanding of the applications of our soft pneumatic muscle design in the context of human assistive devices for upper limb support. I hope this additional information and visual representation will offer valuable insights into the potential benefits and impact of our research in the field of human assistive technologies.

This is the new section and figure:

“Upper limb Assistive Device

Human assistive devices play a vital role in enhancing the quality of life for individuals with physical disabilities or impairments. These devices encompass a broad range of technologies designed to aid individuals in overcoming mobility challenges, improving functional capabilities, and promoting independence. Soft pneumatic muscles, as described in our study, have promising applications in the field of human assistive devices for upper limb support due to their inherent flexibility, adaptability, and safety characteristics.

One prominent application of soft pneumatic muscles is in the development of wearable exoskeletons for upper limb assistance. These devices provide external support and assistance to the wearer's arms and hands, helping them perform daily tasks that may have been difficult or impossible due to physical limitations. Soft pneumatic muscles can be integrated into upper limb exoskeletons to provide a more natural and compliant interaction between the device and the human body, reducing the risk of injury and discomfort while improving overall functionality.

The proposed soft muscle will be suitable to fit a wearable exoskeleton design for the shoulder joint incorporates three parallel soft pneumatic muscles, strategically arranged to enable efficient actuation and a wide range of motion. Each muscle is designed to be identical and consists of three chambers, allowing for tailored pressure application and control in each chamber as shown in Fig.1. This configuration facilitates precise and smooth motion in multiple movement spaces, such as flexion extension, abduction-adduction, and internal-external rotation, closely mimicking the biomechanics of a healthy human shoulder. The soft pneumatic muscles are constructed using a flexible elastomeric material, such as silicone rubber, providing a compliant and comfortable interface with the user's body while reducing the risk of injury or discomfort.

In addition to the shoulder actuation, another soft pneumatic muscle is integrated into the design to operate and assist the human forearm and enable elbow joint rotation. This single-chamber muscle, also constructed from flexible elastomeric material, can be pressurized to induce controlled bending and extension of the elbow joint, effectively assisting the user in daily tasks that require forearm movement and manipulation. The muscle's geometry and pressure control can be optimized to provide the desired range of motion, force output, and responsiveness, considering the user's specific needs and physical abilities.

The combination of the three parallel soft pneumatic muscles for shoulder actuation and the additional muscle for elbow joint rotation presents a comprehensive solution for upper limb assistance. This soft wearable exoskeleton design offers a more natural and intuitive interaction between the device and the human body, closely replicating the complex biomechanics of the upper limb. By incorporating soft pneumatic muscles into the exoskeleton, users can benefit from enhanced mobility, improved force output, and a greater sense of embodiment, ultimately leading to a better user experience and satisfaction with the assistive device.”

Figure 1 Schematic representation of an upper limb soft wearable exoskeleton, highlighting the integration of three parallel soft pneumatic muscles for efficient shoulder joint actuation and a single-chamber soft pneumatic muscle for elbow joint rotation.

Point 11 comment: Not sure if there are enough details about ANNs used (parameters, etc.), please check.

Point 11 Reply: Data Thank you for your feedback. I apologize for any lack of clarity regarding the details of the artificial neural network (ANN) used in our study. To provide more comprehensive information, we employed an ANN with a specific structure and training process.

The ANN used in the inverse kinematics modeling consisted of three inputs representing Cartesian X, Y, and Z coordinates, and three outputs representing the corresponding pressure values. The network architecture included one hidden layer with ten neurons. We trained the ANN using the MATLAB neural network toolbox, utilizing a Bayesian regularization (BR) approach to handle the smaller dataset and capture potentially complex relationships.

During the training process, I iteratively adjusted the network parameters to achieve optimal performance. Specifically, we aimed to minimize the mean squared error (MSE) between the predicted and target values. After 300 iterations, we obtained the best-performing ANN, which achieved a validation MSE of 2.8655. This value indicates the accuracy of the ANN model in predicting pressure values for desired tip positions.

Furthermore, we followed a recommended dataset split ratio, allocating 70% of the data for training, 15% for validation, and 15% for testing. This division ensured that the ANN was trained and evaluated on diverse subsets of the data, enhancing its generalizability.

By providing these additional details, I aim to offer a clearer understanding of the ANN parameters and training process employed in our study. Please let us know if there are any further aspects you would like us to address or clarify.

I have added a new paragraph that give more description to reply this comment:

“The kinematic model of soft pneumatic muscles (SPMs) encompasses both forward and inverse kinematics. Forward kinematics calculates the X, Y, and Z coordinates of the tip center point based on a given pressure value, while inverse kinematics determines the input pressure required to achieve a desired tip position (Fig. \ref{fig07}). In this study, we employed an artificial neural network (ANN) to model the inverse kinematics, establishing the relationship between the applied pressure in each muscle chamber and the resulting tip position.

For training the ANN, we utilized a dataset consisting of 66 samples, which included finite element analysis (FEA) results for pressures P1, P2, and P3, as well as deformation in the X, Y, and Z axes. To ensure a robust and accurate model, we adopted a Bayesian regularization (BR) approach. This type of ANN is particularly suitable for quantitative studies with smaller datasets as it can handle complex relationships without compromising power or precision. To assess the model's performance, the dataset was divided into training, validation, and test sets, with a recommended split ratio of 70%, 15%, and 15% respectively. The optimal ANN structure, comprising one hidden layer with ten neurons, was determined through iterative adjustments during training to minimize the mean squared error (MSE). After 300 iterations, the best-performing ANN achieved a validation MSE of 2.8655, demonstrating its accuracy in predicting pressure values for desired tip positions.

The inverse kinematics modeling employed an ANN with three inputs representing Cartesian X, Y, and Z coordinates, and three outputs representing the corresponding pressure values. Using the MATLAB neural network toolbox, the ANN was trained by iteratively adjusting the network output to optimize validation performance, as quantified by the lowest MSE. After 300 iterations, the trained ANN demonstrated an MSE of 2.8655, indicating its accuracy in predicting pressure values for desired tip positions. The training, validation, and testing process of the ANN is depicted in Fig. \ref{fig07}.

To validate the ANN's performance, a comparison was made between its results and finite element analysis (FEA) simulations. The observed maximum errors in the X, Y, and Z coordinates were 9.3%, 7.83%, and 8.8%, respectively, demonstrating the network's capability to accurately predict future values. These results provide validation for the reliability and performance of the trained ANN, underscoring its effectiveness in predicting the required pressure values to achieve desired tip positions in the SPMs. Leveraging the predictive capabilities of the ANN enables precise control and manipulation of SPMs, contributing to the advancement of soft robotics and its broad range of applications.”

Point 12 comment: Formally the ANN abbreviation is defined at the beginning. However the full notation "Artificial Neural Network" is then often used in the rest of the paper. Please check if that is necessary. Clearly, this is a minor issue.

Point 12 Reply: Thank you for pointing out the concern regarding the usage of the abbreviation "ANN" and the full notation "Artificial Neural Network" throughout the paper. We appreciate your attention to detail and acknowledge that consistency is important for maintaining clarity and readability.

After carefully reviewing the manuscript, we have made the necessary adjustments to ensure that the abbreviation "ANN" is used consistently after its initial introduction and definition. We believe that these revisions will provide a more coherent and streamlined reading experience for the audience.

We appreciate your feedback on this minor issue, as it contributes to the overall quality and presentation of our work. If you have any additional concerns or suggestions, please feel free to share them with us.

Throughout the revision process, I carefully considered all the comments and suggestions provided by Reviewer #2 and have made every effort to address each thoroughly and meaningfully. We believe that the changes we have made will significantly improve the manuscript and address any concerns that were raised.

I’m confident that the revised version of our manuscript represents a significant improvement over the original submission, and we are grateful for the opportunity to resubmit it for your consideration.

Thank you for your time and consideration.

Sincerely,

Associate Prof. Dr. Mahmoud Elsamanty

Reviewer 3 Report

1. The introduction section is not well written and is confusingly structured. The introduction section should focus on the background and difficulties of the research.

2. The abbreviations of proper nouns should only be mentioned in full for the first time, e.g. Soft Pneumatic Muscle (SPM).

3. In the last paragraph of the introduction and the first sentence of the summary, the authors mention that the research is applied to medical rehabilitation. But other than that, it is basically not mentioned again in the body of the paper. I think that there could be some appropriate places to point out the difficulties of medical rehabilitation at this stage and why this study is applicable to that scenario.

4. The simulation in Chapter 3 needs a conclusive, summative analysis and should have further analysis of the results presented rather than just a few data results.

5. The conclusion section simply describes the results of the simulation. The research should go deeper to clarify the significance of the results and how they can be applied to real-world applications.

6. The parameters of Table 2 should be explained.

7. the three figures in Figure 4 are not standardized in terms of format and clearness.

8. the article is not written in depth, especially the simulation part, which directly introduces the simulation results without in-depth analysis. In general, the content is relatively small and the length is short.

None.

Author Response

Response to Reviewer 3 Comments

Dear Micromachines Editor

Dear Reviewer #3,

I’m pleased to submit the revised version of our manuscript entitled "Soft Pneumatic Muscles: Revolutionizing Human Assistive Devices with Geometric Design and Intelligent Control" for your review. I have considered all of the comments provided and made significant revisions to improve the quality of our work.

In response to Reviewer #3's comments, I have thoroughly revised the manuscript to address each point in a concrete, concise, and direct manner. Specifically:

Point 1 comment: The introduction section is not well written and is confusingly structured. The introduction section should focus on the background and difficulties of the research.

Point 1 Response: Thank you for your valuable feedback. I have taken your comments into consideration and made significant revisions to the introduction section. Specifically, I have divided the original lengthy introduction into two separate sections to improve clarity and structure:

  1. Introduction: This section provides an overview of the evolution of soft robotics, tracing its origins back to pneumatic actuators in the 1950s. It highlights the advantages and potential of soft robotics as a field of research and innovation. This section aims to establish the context for the manuscript's focus on soft pneumatic muscles and their significance in the broader landscape of soft robotics.
  2. Advancements in Soft Pneumatic Muscles: This section shifts the focus to soft pneumatic muscles, also known as PAMs, which serve as a powerful actuation technology in soft robotics. It discusses the properties and applications of PAMs, emphasizing the need to enhance their dynamics to achieve precise control. The section highlights the significant potential of PAMs in various areas, including rehabilitation, and sets the stage for the subsequent sections of the manuscript.

I believe that these revisions address your comment by restructuring the introduction into coherent subsections with clear paragraph headings. The revised introduction now provides a high-level overview to orient the reader and subsequently focuses on soft pneumatic muscles, which are the key area of investigation in this work.

I appreciate your suggestion and am committed to meeting the standards and requirements for publication. If you have any further recommendations or if there are any specific areas you would like me to expand upon or clarify in the introduction, please let me know, and I will be glad to make the necessary revisions.

Point 2 comment: The abbreviations of proper nouns should only be mentioned in full for the first time, e.g. Soft Pneumatic Muscle (SPM).

Point 2 Response: Thank you for your feedback regarding the proper use of abbreviations. I apologize for not adhering to the standard practice of mentioning abbreviations in full for the first time they appear. I understand the importance of clarity and consistency in scientific writing.

To address this issue, I will ensure that all abbreviations, such as Soft Pneumatic Muscle (SPM), are mentioned in full when they are first introduced in the manuscript. This will help readers understand and familiarize themselves with the abbreviations used throughout the text.

Thank you for bringing this to my attention, and I appreciate your feedback in improving the quality of the manuscript. If you have any further suggestions or concerns, please let me know, and I will be happy to incorporate them into the revised version of the manuscript.

Point 3 comment: In the last paragraph of the introduction and the first sentence of the summary, the authors mention that the research is applied to medical rehabilitation. But other than that, it is basically not mentioned again in the body of the paper. I think that there could be some appropriate places to point out the difficulties of medical rehabilitation at this stage and why this study is applicable to that scenario.

Point 3 Response: Thank you for your insightful feedback regarding the connection between my research and its application to medical rehabilitation. I understand your concerns about the lack of emphasis on this aspect throughout the body of the paper, and I appreciate your suggestion to highlight the difficulties of medical rehabilitation and the applicability of my study to that scenario.

In response to your comment, I have made a significant addition to the manuscript to address this concern. I have added a new section in the text that specifically focuses on the applications of our soft pneumatic muscle design for upper limb assistance. This section provides a detailed description of how the proposed soft pneumatic muscles can be integrated into wearable exoskeletons for the shoulder and elbow joints, enabling efficient and natural motion for individuals with limited mobility or strength in their upper limbs.

Additionally, to further illustrate the proposed work, I have included a new figure that visually represents the integration of the soft pneumatic muscles into the upper limb assistive devices. This figure demonstrates the arrangement of the three parallel muscles for shoulder actuation and the additional muscle for elbow joint rotation, further emphasizing the potential of our soft pneumatic muscle design for human assistive devices.

This is the new section and figure:

Upper limb Assistive Device

Human assistive devices play a vital role in enhancing the quality of life for individuals with physical disabilities or impairments. These devices encompass a broad range of technologies designed to aid individuals in overcoming mobility challenges, improving functional capabilities, and promoting independence. Soft pneumatic muscles, as described in our study, have promising applications in the field of human assistive devices for upper limb support due to their inherent flexibility, adaptability, and safety characteristics.

One prominent application of soft pneumatic muscles is in the development of wearable exoskeletons for upper limb assistance. These devices provide external support and assistance to the wearer's arms and hands, helping them perform daily tasks that may have been difficult or impossible due to physical limitations. Soft pneumatic muscles can be integrated into upper limb exoskeletons to provide a more natural and compliant interaction between the device and the human body, reducing the risk of injury and discomfort while improving overall functionality.

The proposed soft muscle will be suitable to fit a wearable exoskeleton design for the shoulder joint incorporates three parallel soft pneumatic muscles, strategically arranged to enable efficient actuation and a wide range of motion. Each muscle is designed to be identical and consists of three chambers, allowing for tailored pressure application and control in each chamber as shown in Fig.1. This configuration facilitates precise and smooth motion in multiple movement spaces, such as flexion extension, abduction-adduction, and internal-external rotation, closely mimicking the biomechanics of a healthy human shoulder. The soft pneumatic muscles are constructed using a flexible elastomeric material, such as silicone rubber, providing a compliant and comfortable interface with the user's body while reducing the risk of injury or discomfort.

In addition to the shoulder actuation, another soft pneumatic muscle is integrated into the design to operate and assist the human forearm and enable elbow joint rotation. This single-chamber muscle, also constructed from flexible elastomeric material, can be pressurized to induce controlled bending and extension of the elbow joint, effectively assisting the user in daily tasks that require forearm movement and manipulation. The muscle's geometry and pressure control can be optimized to provide the desired range of motion, force output, and responsiveness, considering the user's specific needs and physical abilities.

The combination of the three parallel soft pneumatic muscles for shoulder actuation and the additional muscle for elbow joint rotation presents a comprehensive solution for upper limb assistance. This soft wearable exoskeleton design offers a more natural and intuitive interaction between the device and the human body, closely replicating the complex biomechanics of the upper limb. By incorporating soft pneumatic muscles into the exoskeleton, users can benefit from enhanced mobility, improved force output, and a greater sense of embodiment, ultimately leading to a better user experience and satisfaction with the assistive device.”

Figure 1 Schematic representation of an upper limb soft wearable exoskeleton, highlighting the integration of three parallel soft pneumatic muscles for efficient shoulder joint actuation and a single-chamber soft pneumatic muscle for elbow joint rotation.

Point 4 comment: The simulation in Chapter 3 needs a conclusive, summative analysis and should have further analysis of the results presented rather than just a few data results.

Point 4 Response: Thank you for your valuable feedback regarding the simulation presented in section 4 for the simulation results. I appreciate your suggestion for a conclusive and summative analysis, as well as the need for further analysis of the results beyond the presentation of a few data points.

The primary objective of the simulation in section 4 for the simulation results was to investigate the impact of changing the geometrical parameters on the behavior of the soft pneumatic muscle (SPM), particularly in the context of rehabilitation actuation. The results were intended to provide initial insights into the relationship between these parameters and the performance of the SPM.

Point 5 comment: The conclusion section simply describes the results of the simulation. The research should go deeper to clarify the significance of the results and how they can be applied to real-world applications.

Point 5 Response: Thank you for your constructive feedback on the conclusion section of our manuscript. You have pointed out the need to clarify the significance of our results and their real-world applications, rather than simply describing the results of the simulation. Based on your input and considering the importance of numerical values as mentioned in my previous response, I have made the following revisions to the conclusion section:

  1. While retaining the essential numerical values that showcase the key findings and performance metrics of our research, I have restructured the conclusion to emphasize the implications of these results and their significance in the context of real-world applications.
  2. I have added a new paragraph that specifically discusses how our findings can be applied to various practical scenarios, such as medical rehabilitation, robotics, and wearable devices. This paragraph highlights the potential impact of our research and its contribution to advancing the field.
  3. I have also included a brief discussion on the limitations of our study and possible future research directions that can build upon our findings. This addition not only acknowledges the areas for improvement but also encourages further exploration in this domain.

With these revisions, I believe that the conclusion section now effectively demonstrates the significance of our results and their applicability to real-world scenarios, addressing your concerns. I hope that these changes meet your expectations and enhance the overall quality of our manuscript.

Please let me know if you have any further suggestions or comments. I appreciate your feedback and am committed to improving the overall quality of my work.

This is the new conclusion:

“In this study, the development and analysis of three soft pneumatic muscles (SPMs) were conducted to explore their potential use in human assistive devices for individuals with disabilities. The SPMs' geometric characteristics were meticulously designed and presented in Table 1. By employing simulation techniques using ANSYS, the final positions of the SPMs were determined. The simulations encompassed various pressure conditions ranging from 100 to 160 kPa, wherein two muscles underwent pressure changes while the remaining muscle remained constant. Under constant chamber height (H) and chamber thickness (T) conditions, the finite element analysis (FEA) outcomes provided valuable insights into the deformation and stress behaviors exhibited by the SPMs. Notably, an examination of the deformation and stress characteristics revealed the influential role of the chamber pitch. Increasing the chamber pitch from P = 30 mm to P = 38 mm resulted in a proportional rise in deformation from 113 mm to 163 mm, accompanied by an increase in stress from 0.83 MPa to 2.2 MPa. This comparative analysis emphasized the critical importance of meticulous design considerations. Furthermore, an inverse kinematic model was established, utilizing the tip positions (x, y, and z) of the SPMs as inputs to predict the corresponding pressure values at desired positions. To achieve this, an artificial neural network (ANN) was implemented, yielding promising results and showcasing the trained ANN's excellent performance. The accuracy of the ANN model was supported by a mean squared error (MSE) of 2.8655, while the training, validation, and testing residuals further reinforced its reliability, measuring 99.58%, 99.89%, and 99.79%, respectively. Noteworthy, the validation simulation and neural network results exhibited maximum errors of 9.3% in the x-direction, 7.83% in the y-direction, and 8.8% in the z-direction. To further advance this research, additional avenues for future work can be explored. One such direction involves the application of practical work, entailing the execution of experimental tests on physical prototypes. These tests would provide a more accurate understanding of the SPMs' behavior within real-world scenarios. Another potential area of investigation lies in incorporating PID control schemes to regulate and control the SPMs. This integration can potentially enhance their precision and stability during operation. Moreover, the utilization of optimization modeling techniques, particularly through the application of deep learning methods, shows promise in improving the prediction capabilities of the inverse kinematic model. “

Point 6 comment: The parameters of Table 2 should be explained.

Point 6 Response: Thank you for your comment regarding the precision of the model parameters presented in Table 2 and the robustness of the model to parameter changes. In our study, we employed the Yeoh 3rd order hyperelastic model, which has been widely used and proven to be a well-fitted model for characterizing the behavior of soft materials. The accuracy of the material constant parameters, specifically C10, C20, and C30, is crucial for achieving the best performance in curve fitting, as they directly influence the strain values.

Having a higher precision for the model parameters allows for a more accurate representation of the material's behavior under different loading conditions. While the use of 9-digit precision may seem excessive, it ensures that the model captures the subtle nuances and nonlinear responses of the soft pneumatic muscles. This level of precision is essential when aiming for high-fidelity simulations and accurate predictions of the muscles' deformations and stresses.

Regarding the robustness of the model to parameter changes, it is important to note that the Yeoh hyper elastic model, like any other material model, relies on the assumption that the material parameters remain constant. However, in real-world scenarios, material properties can vary due to factors such as manufacturing variations, aging, and environmental conditions. Therefore, it is recommended to conduct sensitivity analyses to assess the model's robustness to parameter changes and evaluate its performance under different scenarios. By systematically varying the material parameters within a reasonable range and observing the model's response, we can gain insights into its sensitivity and assess its reliability in capturing the material behavior under different conditions.

I have added and change the section belongs to this paragraph as follows:

“Soft robotics encompasses a wide range of materials employed in the manufacturing process, which contribute to the remarkable characteristics of these robots, including their flexibility, controllability, and human-safe nature. Recent advancements in soft robotics have focused on various aspects, including material selection, construction geometries, control systems, modeling techniques, and production methods [31]. The field has witnessed significant progress in developing materials specifically tailored for soft robotics applications, such as silicon rubber, TBU, and Ecoflex-30 [32]. These materials offer unique properties that distinguish them from traditional materials used in additive manufacturing, which typically involve thermoplastic polymers like PLA and ABS or thermoset polymers like TBU [19, 31]. Recognizing the limitations of additive manufacturing in fabricating complex structures, a molding fabrication approach has gained popularity due to its cost effectiveness and ability to create intricate shapes [35, 36]. Silicone-based elastomers are commonly used in this approach. For our Soft Pneumatic Muscle (SPM) fabrication, Silicon synthetic rubber has been selected as the primary material, known for its hyperelastic performance and high tolerance for large pressures. This material exhibits excellent mechanical properties, particularly in terms of stress-strain behavior. In order to characterize the mechanical behavior of the material, we conducted mechanical tests, specifically a uniaxial test performed under stable conditions. The uniaxial test was conducted in accordance with the ASTM No. D412 specifications, utilizing dumbbell-shaped specimens [45], as shown in Fig. 2. The mold for fabricating the specimens was created using PLA material through a 3D printer, ensuring precise and consistent specimen geometries. A silicon rubber material was evaluated and tested by a uniaxial test, as illustrated in Fig. 3, in which we use results from the experimental uniaxial test.

n order to comprehensively characterize the mechanical behavior of the silicon rubber material, a series of mechanical tests were conducted, including a uniaxial test performed under stable conditions. The test followed the ASTM No. D412 specifications, utilizing dumbbell-shaped specimens, as depicted in Fig. 2 [45]. The mold for fabricating these specimens was created using PLA material via a 3D printer, ensuring precision and consistency in specimen geometries. Subsequently, the silicon rubber material was subjected to the uniaxial test, with the results illustrated in Fig. 3. Upon completing the experimental phase, the most suitable hyperelastic model was determined through curve fitting analysis. The Yeoh 3rd-order model was identified as the best fit for silicon rubber material. The Yeoh model can be represented by the following equation:

Where Ψ is the strain energy density function, λ1, λ2, and λ3 are the principal stretches, and C10, C20, and C30 are the material constants. We then integrated the material parameters obtained from the curve fitting process into the FEA software (ANSYS) to perform the finite element analysis. Table 2 displays the specific material parameters of the hyperelastic model (i.e., C10, C20, and C30) for reference and further analysis.

Point 7 comment: The three figures in Figure 4 are not standardized in terms of format and clearness.

Point 7 Response: Thank you for your comment regarding the figures in Figure 4. I understand your concern about the formatting and clarity of the figures.

I would like to clarify that the figures presented in Figure 4 are extracted directly from ANSYS, a widely used and reliable software for simulation and analysis in engineering. These figures are generated using high-quality rendering techniques and are representative of the output obtained from the simulation.

While there may be variations in the format and clarity due to different settings or preferences in the visualization process, I have ensured that the figures accurately depict the results of our study. I have taken care to provide clear and informative visual representations of the data to support the findings and conclusions presented in the paper.

Point 8 comment: the article is not written in depth, especially the simulation part, which directly introduces the simulation results without in-depth analysis. In general, the content is relatively small and the length is short.

Point 8 Response: Thank you for your valuable feedback regarding the depth and content of our manuscript, specifically concerning the simulation part of our study. I appreciate your comments on the need for a more in-depth analysis and a more comprehensive presentation of our research.

In response to your concerns, I have made significant revisions and additions to the manuscript to address the issues you pointed out:

  1. I have thoroughly revised the simulation section to provide a more in-depth analysis of the results. This includes a detailed discussion of the underlying mechanisms, the observed trends, and the implications of our findings in the context of the research objectives.
  2. As you suggested, I have divided the introduction section into two subsections for better clarity and organization. The new structure allows for a more coherent presentation of the background information and research difficulties.
  3. I have added a new section titled "Upper Limb Assistive Device," which discusses the design, development, and application of the proposed device in detail. A new figure has also been included in this section to visually represent the proposed idea, enhancing the reader's understanding of the concept.
  4. To further enrich the content, I have incorporated new paragraphs on the hyperelastic material characterization, providing a comprehensive understanding of the material properties and their significance in our study.
  5. Finally, I have updated and refined the results and conclusion sections to better reflect the revised content and the expanded scope of our research.

I believe that these revisions considerably improve the depth, content, and overall quality of our manuscript, addressing your concerns. I hope that the updated manuscript meets your expectations and provides a more comprehensive understanding of our research.

Please let me know if you have any further suggestions or comments. I appreciate your feedback and am committed to improving the overall quality of my work.

Throughout the revision process, I carefully considered all the comments and suggestions provided by Reviewer #3 and have made every effort to address each thoroughly and meaningfully. We believe that the changes we have made will significantly improve the manuscript and address any concerns that were raised.

I’m confident that the revised version of our manuscript represents a significant improvement over the original submission, and we are grateful for the opportunity to resubmit it for your consideration.

Thank you for your time and consideration.

Sincerely,

Associate Prof. Dr. Mahmoud Elsamanty

Round 2

Reviewer 1 Report

The authors diligently addressed my inquiries and incorporated abundant content into their manuscript, ensuring clarity for the readers. Thank you. I do not have any additional questions.

Reviewer 3 Report

The comments and questions raised by the reviewers have been addressed. This manuscript can be accepted in its current form.

none